# Serine Metabolism in Health and Disease and as a Conditionally Essential Amino Acid

**DOI:** 10.3390/nu14091987

**Published:** 2022-05-09

**Authors:** Milan Holeček

**Affiliations:** Department of Physiology, Faculty of Medicine in Hradec Králové, Charles University, Šimkova 870, 500 03 Hradec Králové, Czech Republic; holecek@lfhk.cuni.cz

**Keywords:** diabetes, serine supplementation, neuropathy, glycine, deoxysphingolipids, hyperhocysteinemia

## Abstract

L-serine plays an essential role in a broad range of cellular functions including protein synthesis, neurotransmission, and folate and methionine cycles and synthesis of sphingolipids, phospholipids, and sulphur containing amino acids. A hydroxyl side-chain of L-serine contributes to polarity of proteins, and serves as a primary site for binding a phosphate group to regulate protein function. D-serine, its D-isoform, has a unique role. Recent studies indicate increased requirements for L-serine and its potential therapeutic use in some diseases. L-serine deficiency is associated with impaired function of the nervous system, primarily due to abnormal metabolism of phospholipids and sphingolipids, particularly increased synthesis of deoxysphingolipids. Therapeutic benefits of L-serine have been reported in primary disorders of serine metabolism, diabetic neuropathy, hyperhomocysteinemia, and amyotrophic lateral sclerosis. Use of L-serine and its metabolic products, specifically D-serine and phosphatidylserine, has been investigated for the therapy of renal diseases, central nervous system injury, and in a wide range of neurological and psychiatric disorders. It is concluded that there are disorders in which humans cannot synthesize L-serine in sufficient quantities, that L-serine is effective in therapy of disorders associated with its deficiency, and that L-serine should be classified as a “conditionally essential” amino acid.

## 1. Introduction

In humans, L-serine can be synthesized from 3-phosphoglycerate (3-PG) and glycine. Therefore, it is classified as a nutritionally non-essential (dispensable) amino acid in textbooks of biochemistry and physiology. L-serine has an exceptional importance in metabolism of all nutrients and in a broad range of cellular functions. L-serine is a substrate for glucose and protein synthesis, and it is a building block of phospholipids, particularly phosphatidylserine (PS), and sphingolipids (SL), such as ceramides, phosphosphingolipids, and glycosphingolipids, which are highly concentrated in all cell membranes. L-serine is important also in the synthesis of glycine and in folate and methionine cycles, in synthesis of sulphur-containing amino acids, and in neurotransmission. A hydroxyl side-chain of L-serine contributes to the polarity of proteins and formation of glycoproteins, and serves as a primary site for binding a phosphate group to regulate protein function. Its D-isoform, D-serine, has unique metabolic significance (Figure 1).

Recent articles demonstrate that L-serine deficiency is associated with abnormal metabolism of phospholipids and sphingolipids, and impaired development and function of the nervous system [1,2,3,4,5] and several studies have reported benefits of L-serine in therapy of neurological problems due to primary defects of serine biosynthesis, hereditary sensory neuropathy type 1, amyotrophic lateral sclerosis, and diabetes [2,4,6,7,8,9]. Currently, L-serine and its metabolic products, specifically D-serine and phosphatidylserine, are being investigated for the therapy of hyperhocysteinemia and a wide range of neurological and psychiatric disorders including Alzheimer’s disease, Parkinson’s disease, Huntington’s disease, and schizophrenia. 

The first part of the article provides an overview of the physiological importance of L-serine and pathways of its metabolism. It will then focus on alterations in serine metabolism in some diseases, its therapeutic potential, and the suggestion that L-serine should be classified as a conditionally essential amino acid (CEAA). The conceptual framework of the article is shown in Figure 2.

## 2. Sources of Serine

The main sources of serine for humans are dietary intake, degradation of endogenous proteins, and de novo synthesis from 3-PG and glycine.

### 2.1. Dietary Sources 

The main dietary source of L-serine is protein, in which L-serine content ranges between 2 and 5% [10]. Lipids, that contain serine in the form of PS and SL, are much less important because the content of PS in the diet is very low and most SL are utilized by the intestinal mucosa or excreted in the faces [11]. Hence, at the daily intake of ~1 g protein per kg of body weight, the amount of serine obtained from food ranges between 1.4 and 3.5 g (13.2–33.0 mmol) per day in an adult.

### 2.2. Degradation of Endogenous Proteins

Assuming that serine content in human muscle protein is ~2.3% [10] and that skeletal muscle plays a major role in protein turnover, which in a 70 kg individual is about 300 g per day, approximately 6.9 g/day (65.1 mmol/day) of L-serine is released during degradation of endogenous proteins. This estimation is in excellent agreement with measurement of the rate of serine appearance following an overnight fast (~58 mmol/day) by use of stable isotope tracers [12].

### 2.3. Synthesis from 3-PG and Glycine

It has been estimated that ~73% of the serine appearance rate in fasting humans is the result of serine synthesis from 3-PG and glycine [12].

#### 2.3.1. Synthesis from 3-PG 

The first step in L-serine synthesis from 3-PG, which is formed during glycolysis or gluconeogenesis, is oxidation of 3-PG by 3-PG dehydrogenase to 3-phosphohydroxypyruvate. The second step is transamination with glutamate to form 3-phosphoserine catalyzed by 3-phosphoserine aminotransferase. The final step is irreversible hydrolysis to form L-serine and inorganic phosphate by phosphoserine phosphatase. The final step is considered as the rate limiting step and is subject to feedback inhibition of L-serine synthesis [13]. L-serine synthesis from 3-PG is high in the brain (especially in astrocytes) and the kidneys. L-serine biosynthetic enzymes in the liver are activated by protein restriction or carbohydrate-rich diet [12,14,15].

#### 2.3.2. Synthesis from Glycine

The enzyme serine hydroxymethyltransferase catalyzes the reversible conversion of glycine to serine by the transfer of one-carbon from 5,10-methylene-tetrahydrofolate (5,10-MTHF) to glycine forming tetrahydrofolate (THF) and L-serine. In healthy humans, L-serine synthesis by serine hydroxymethyltransferase accounts for approximately 41% of the whole-body glycine flux [16].

#### 2.3.3. Role of the Kidneys in Serine Synthesis

The measurements of arterial–venous differences have shown that the kidneys are the main source of L-serine released to the blood stream during fasting and that under physiological conditions the kidneys produce about 4 g (~40 mmol) of L-serine per day [12,17]. The kidneys synthesize L-serine in the cells of the proximal tubules both from 3-PG and glycine [18]. The main sources of 3-PG are gluconeogenic precursors, such as pyruvate, lactate, glutamate, glutamine, and aspartate. The role of gluconeogenesis has been confirmed by a marked decrease in serine efflux from the kidneys after inhibition of phosphoenolpyruvate carboxykinase (the key enzyme of gluconeogenesis) by 3-mercaptopicolinate [19]. The conversion of glycine to serine is mediated by the combined action of the glycine cleavage enzyme and serine hydroxymethyltransferase [20]. The pathway is activated during acidosis and serves as one of the sources of ammonia for the elimination of H^+^ from the body by urine (Figure 3).

## 3. Serine Degradation

Serine is a glucogenic amino acid, and can contribute to gluconeogenesis by two pathways (Figure 4). In the first, L-serine is converted directly to pyruvate by the action of serine dehydratase. This pathway is activated in the liver by increasing the protein content of the diet [21]. In the second pathway, serine is converted by serine-pyruvate transaminase, glycerate dehydrogenase, and glycerate kinase to 2-phosphoglycerate, which may enter the pathways of glycolysis and gluconeogenesis.

The major routes of serine degradation in humans are via glycine and transsulfuration pathway. Glycine, produced from L-serine in reaction catalyzed by glycine hydroxymethyltransferase, is degraded by a group of enzymes called the glycine cleavage system to yield carbon dioxide, ammonia, and 5,10-MTHF. A less significant possibility of glycine degradation is deamination by D-aminooxidase to glyoxylate, which is oxidized to oxalate to be excreted in the urine. The transsulfuration pathway, in which L-serine acts as a substrate for synthesis of cystathionine, is described in Section 5.3.

## 4. Transporters

Serine is transported across the plasma membranes by any of the transporters for small neutral amino acids:System A (alanine preferring)—a sodium-dependent transporter for small neutral amino acids. It is found in most cell types of the body and includes transporters SNAT1 (SLC38A1), SNAT2 (SLC38A2), and SNAT4 (SLC38A4). Transporter recruitment from vesicles under the plasma membrane to the cell surface is sensitive to amino acid levels and hormones (notably to insulin and glucagon). The system plays a role in tissue amino acid uptake after food intake and in hepatic amino acid uptake to be used for gluconeogenesis during starvation [22,23,24].System ASC (alanine, serine, and cysteine preferring)— a sodium-dependent transporter, which includes transporters ASCT1 (SLC1A4) and ASCT2 (SLC1A5). It is supposed that ASCT1 is the main transporter for serine in the brain, where it plays a key role in transporting serine from astrocytes and serine uptake by neurons [25]. ASCT1 mutations cause a disorder of intellectual disability, progressive microcephaly, spasticity and thin corpus callosum [26].System asc—a heterodimeric (SLC7A10/SLC3A2) sodium-independent amino acid exchanger. The system includes transporter Asc-1, which is expressed in the brain and has a high affinity for both L- and D-serine [27].

## 5. Physiological Functions 

The physiological importance of serine is exceptional and there is no other amino acid that can match its range of functions, some of which are mediated by glycine (Figure 5).

### 5.1. L-Serine and Proteins

The L-serine content in most proteins, including skeletal muscle, collagen and milk, ranges between 2 and 5%. Due to its polar side chain with a hydroxyl group, it is located predominantly on the surface of proteins, where it contributes to hydrophilicity and interactions of proteins with other substances. L-serine residues are a part of the O-glycosidic bonds of glycoproteins and serve as a primary site to reversibly bind a phosphate group to regulate protein function. L-serine is found in the catalytic sites of ~200 hydrolytic enzymes, the so-called serine hydrolases, including trypsin, chymotrypsin, and lipoprotein and hormone sensitive lipase. High amounts of phosphorus bound to the protein by a serine ester linkage are in casein, the major protein in milk, which is, therefore, classified as a phosphoprotein.

### 5.2. L-Serine and Folate and Methionine Cycles

L-serine is the main one-carbon donor for reversible transfer to tetrahydrofolate (THF) to form glycine and N^5^,N^10^-CH_2_-THF, which can be reduced to N^5^-methyltetrahydrofolate (CH_3_-THF), a substrate for homocysteine methylation to methionine [28]. The entry of L-serine into the folate and methionine cycles enables its role in maintaining the availability of one-carbon groups for several metabolic pathways including histidine catabolism and synthesis of purines and pyrimidines and via S-adenosylmethionine (SAMe) to participate in many methylation reactions, such as methylation of proteins and nucleic acids and synthesis of phosphatidylcholine, creatine, sarcosine, and epinephrine (Figure 5). Since glycine synthesis from L-serine depends on THF availability, THF depletion may cause a net reduction in the capacity of glycine synthesis [29,30].

### 5.3. L-Serine and Transsulfuration Pathway

In the reaction catalyzed by cystathionine β-synthase, L-serine initiates the transsulfuration pathway, which is connected to the methionine cycle by its intermediate homocysteine. This makes L-serine important for both possible routes of homocysteine disposal (the first is homocysteine methylation to methionine mentioned in the previous section) and synthesis of several sulfur-containing substances, such as cysteine, cystine, taurine, and glutathione (Figure 6).

### 5.4. L-Serine and Sphingolipids (SL)

The first and rate-limiting step in de novo synthesis of SL (ceramides, phosphosphingolipids and glycosphingolipids) is condensation of fatty acyl-CoA (preferred substrate is palmitic acid) and L-serine to form 3-ketosphinganine (3-keto-dihydrosphingosine) in a reaction catalyzed by serine palmitoyltransferase. 3-Ketosphinganine is reduced to sphinganine (dihydrosphingosine), which may be desaturated to sphingenine (sphingosine). Sphinganine and sphingenine are recognized as the fundamental building blocks of all sphingolipids, which may be acylated by one of several isoforms of ceramide synthase with different specificity for acyl-CoA substrates to form dihydroceramides and ceramides (Figure 7).

Ceramides and dihydroceramides are substrates for synthesis of complex SL, sphingomyelins and glykosphingolipids, which are present in large amounts in white matter of the brain and in the myelin sheaths of nerves and play an important role in signal transduction and brain development [31]. Sphingomyelins (phosphosphingolipids) have a phosphocholine or phosphoethanolamine linked to the hydroxyl group of ceramides at the position of a serine side chain. Glykosphingolipids (cerebrosides and gangliosides) are ceramides with one or more sugar residues (glucose or galactose). Cerebrosides contain one molecule of glucose or galactose, gangliosides have several sugars, one of which must be a sialic acid, usually N-acetylneuraminic acid.

In addition to palmitic acid and L-serine, serine palmitoyltransferase can also use other acyl-CoAs and amino acids as substrates, resulting in synthesis of a wide spectrum of sphingoid bases. The use of L-alanine and glycine generates 1-deoxysphinganine and 1-deoxymethylsphinganine, precursors of group 1-deoxysphingolipids (DSLs), which lack the C1 hydroxyl group and therefore cannot be used for synthesis of complex SL. Several studies have shown neurotoxic effects of DSLs and that DSLs cumulate under conditions of decreased L-serine availability [2,4,5,32,33]. It has been suggested that DSLs play an important role in pathogenesis of hereditary disorders of L-serine metabolism and other disorders associated with L-serine deficiency, specifically diabetes and renal failure [2,3,33,34,35].

### 5.5. L-Serine and Phosphatidylserine (PS)

PS consists of two fatty acids attached to the first and second carbon of glycerol and the phosphate linkage at the 3rd position covalently linked to serine. The fatty acid at position 1 is saturated (R1), and the fatty acid at position 2, most commonly docosahexaenoic acid or octadecanoic acid, is unsaturated (R2) (Figure 8). PS is synthesized by two distinct PS synthases that exchange serine for choline in the phosphatidylcholine molecule or ethanolamine in the phosphatidylethanolamine molecule (see Figure 4).

PS plays, like other phospholipids, an important role in membrane fluidity, intercellular communication, and the proper functioning of membrane-bound proteins. Physiologically, high amounts of PS are in the inner part and small amounts are on the outer part of the plasma membrane. During cellular activation and/or apoptosis induction, PS is externalized on the outer surface via the activation of phospholipid scramblases. Externalized PS acts as an immunosuppressive signal that promotes immune tolerance and efferocytosis (the engulfment of apoptotic cells by phagocytes). Pathologically, as in infection, autoimmune diseases, and cancer, the role of PS signaling in cell membranes may be dysregulated [1,36]. PS can be decarboxylated to phosphatidylethanolamine, which is a substrate for three step methylation to phosphatidylcholine and used for synthesis of the fatty acid neurotransmitter anandamide (also known as N-arachidonoylethanolamine) and glycosylphosphatidylinositol anchors of cell-surface proteins [37,38].

Supplementation with phosphatidylserine has been shown to affect cell-surface receptors, signal transduction, and neurotransmission. Clinical trials suggest that PS may have applications for the prevention and therapy of cognitive disorders [39].

### 5.6. Serine and Neurotransmission (D-Serine)

The function of serine in neurotransmission is mediated by L-serine itself and its metabolites, glycine, and D-serine.

#### 5.6.1. L-Serine

L-serine is an agonist of the glycine receptor and, therefore, it is classified as an inhibitory neurotransmitter, which reduces the excitability of neurons, promotes the proliferation of neural stem cells, and has a neuroprotective effect [40,41]. It is the unanimous opinion that L-serine alleviates glutamate neurotoxicity by activating the glycine receptor when the central nervous system is damaged due to ischemia, trauma, and poisoning [40,41,42].

#### 5.6.2. Glycine

Glycine, synthesized from L-serine in the reversible reaction catalyzed by serine hydroxymethyltransferase, induces the opening of channels for chloride ions, hyperpolarization and decreased excitability of postsynaptic neurons. The effect is mainly in the brainstem and spinal cord. In addition, glycine acts as co-agonist of the extrasynaptic N-methyl-D-aspartate (NMDA) subtype of glutamate receptors, which are involved in neurodegenerative disorders and cell death [43,44].

#### 5.6.3. D-Serine

The sources of D-serine are food, intestinal bacteria, degradation of endogenous proteins containing D-serine due to racemization during ageing, and serine racemase, which catalyzes the stereochemical conversion of L-serine to D-serine. In the brain, the main location of D-serine synthesis by racemase is astrocytes, from which it can be released by glutamate through stimulation of AMPA (α-amino-3-hydroxy-5-methylisoxazole-4-propionic acid) receptors [45].

D-serine is a co-agonist of glutamate at the glycine site of the N-methyl-D-aspartate (NMDA) receptor for its full activation [46]. While glycine acts mainly on extrasynaptic NMDA receptors, D-serine gates synaptic NMDA receptors and facilitates long-term potentiation (a persistent strengthening of synapses). Synaptic NMDA receptors are involved in excitatory neurotransmission and long-term synaptic plasticity in the hippocampus and play a fundamental role in memory and learning [47].

D-serine is degraded by D-amino acid oxidase (D-serine + O_2_ → hydroxypyruvate + NH_3_ + H_2_O_2_) or excreted by urine [48]. Serine racemase and D-amino acid oxidase are localized in several tissues, predominantly in the brain and the kidneys. Under physiological conditions, the excretion of D-serine is much greater than that of L-serine due to low renal resorption of D-serine.

D-serine has been proposed for use as an adjunct agent with antipsychotics in the treatment of schizophrenia [49,50]. In addition to nervous tissue, D-serine is currently being investigated in kidneys, cartilage, bones, liver, and enteric nervous system [48].

## 6. Serine and Disease

### 6.1. Primary Disorders of L-Serine Synthesis

The supply of L-serine from the blood stream through the blood–brain barrier is not sufficient to meet the needs of the brain. Therefore, the essential source of L-serine for the brain is its synthesis from 3-PG and the major consequences of primary defects of L-serine synthesis are disorders of brain development and function [51,52]. The disorders can be caused by mutations of any of three genes encoding enzymes of the L-serine synthesis from 3-PG, namely 3-PG dehydrogenase, phosphoserine aminotransferase and phosphoserine phosphatase (enzymes 4–6 in Figure 4).

The disorders are characterized by abnormally low levels of L-serine and glycine in plasma and cerebrospinal fluid, increased levels of DSLs, brain atrophy and hypomyelination. Patients exhibit psychomotor retardation, irritability, growth deficiency, limb and skin defects, epilepsy, and polyneuropathy [51,53]. A lethal anomaly is known as Neu-Laxova syndrome characterized by intrauterine growth restriction, microcephaly, and limb and skin defects. The majority are stillborn or die in the immediate neonatal period [53].

Several studies have demonstrated that L-serine therapy starting soon after birth or during pregnancy may prevent or ameliorate symptoms of serine biosynthesis defects [6,25,54,55].

### 6.2. Hereditary Sensory Neuropathy Type 1

Hereditary sensory neuropathy type 1 is caused by missense mutation of one of three subunits of serine palmitoyltransferase resulting in a shift in the substrate specificity of the enzyme from serine to alanine and glycine and subsequent formation of DSL, 1-deoxysphinganine and 1-deoxymethylsphinganine. Both metabolites accumulate in tissues and exert detrimental effects on neurite formation [2]. The disease is characterized by loss of pain and temperature sensation, and often atrophy and weakness of limb muscles due to degeneration of motor neurons. Oral L-serine supplementation in mice and humans with hereditary sensory autonomic neuropathy type 1 reduced DSL production, and some patients reported an increase in sensation [56].

### 6.3. Disorders of the Nervous System

Neurological abnormalities occurring in patients with primary disorders of L-serine synthesis prove that the de novo synthesis of L-serine plays an essential role in the development and function of the central nervous system. It is supposed that dysregulation of serine metabolism plays a role in pathogenesis of schizophrenia and several neurodegenerative disorders, including Alzheimer’s disease, Parkinson’s disease, amyotrophic lateral sclerosis, and Huntington’s disease. Recently it was shown that glycolysis is impaired in astrocytes in a mouse model of Alzheimer’s disease and that the decrease in glycolysis leads to the reduction of both L- and D-serine synthesis and to the alteration of the synaptic plasticity and memory [57]. Hence, along with new knowledge about the role of serine, efforts are being made to use serine and its derivatives in therapy. Here are some examples:L-serine supplementation has been investigated in the therapy of central nervous system injury, such as that due to cerebral ischemia, stroke, and trauma [40].L-serine has shown therapeutic potential in amyotrophic lateral sclerosis [8].D-serine has been proposed for use as an adjunct agent with antipsychotics in the treatment of schizophrenia [49,50].Clinical trials indicate that PS may prevent and treat depression and age-related cognitive disorders like Alzheimer’s disease [39].

### 6.4. Serine and Diabetes Mellitus

Concentrations of L-serine in plasma and tissues markedly decrease in both type 1 [58,59,60] and type 2 diabetes mellitus [34,61,62,63,64]. The cause is probably impaired glycolysis resulting in decreased synthesis of 3-PG, the precursor for endogenous synthesis of L-serine.

Evidence is growing that L-serine deficiency plays a role in diabetic neuropathy, which may affect both limbs (peripheral neuropathy) and internal organs (autonomic neuropathy), leading to neurological problems and problems with blood pressure, digestive system, sexual organs, sweat glands, and eyes. It is supposed that altered synthesis of SL play a key role, especially increased levels of neurotoxic DSLs [3,4,34,35]. In addition to the role in pathogenesis of neuropathy, it has been shown that DSLs are cytotoxic for pancreatic β-cells, suggesting that their increased levels may contribute to impaired glucose homeostasis and pathogenesis of diabetes itself [5,65]. Recently, DSLs have been suggested as a predictive biomarker for type 2 diabetes mellitus [4,5]. There are several studies reporting that L-serine supplementation reduces DSL concentration and improves glucose homeostasis and signs of neuropathy in diabetes [4,9,32,56]. Decreased L-serine levels may also play a role in elevated homocysteine levels, routinely observed in diabetic patients.

### 6.5. Serine and Kidney Diseases

The role of the kidneys as a source of L-serine for needs of other tissues is undoubtedly the main cause of its decreased plasma concentrations in chronic renal failure [66,67,68,69,70]. Due to the role of L-serine in conversion of homocysteine to methionine and/or cystathionine (Figure 6), decreased L-serine levels may be involved in hyperhomocysteinemia, which is invariably seen in chronic kidney disease [71]. L-serine deficiency may also result in its replacement by alanine or glycine in serine palmitoyltransferase reaction and synthesis of DSL, and play a role in neurologic complications, which occur in every patient with uremic syndrome [72]. Increases in DSL levels were reported recently in a rat 5/6 nephrectomy model of chronic renal insufficiency and in serum from patients with chronic kidney disease [33]. Unfortunately, there are no data on the therapeutic effect of L-serine in renal failure.

#### D-Serine

While plasma concentrations of L-serine in patients with chronic renal disease decrease, concentrations of its enantiomer, D-serine, increase. The cause of this phenomenon is that the kidney proximal tubules reabsorb serine with chiral-selectivity, with D-serine being reabsorbed much less efficiently than L-serine. It has been shown that D-serine levels in the blood strongly correlate with the progression of chronic kidney disease and, therefore, D-serine is emerging as a potential biomarker for kidney disease [73].

It has been known for many years that exogenous administration of D-serine leads to necrosis of proximal tubules [74]. However, the mechanism is not clear and two explanations have been suggested. The first is based on increased production of hydrogen peroxide and other reactive oxygen species due to high activity of D-amino acid oxidase in proximal tubules [75]; the second on hyper-activation of NMDA receptors resulting in vasoconstriction and poor renal perfusion [76].

### 6.6. Serine and Cancer

Currently, there is a significant increase in the interest in the role of L-serine in the pathogenesis of cancer. Recent studies have proven that L-serine synthesizing enzymes, notably 3-PG dehydrogenase, are up-regulated in neoplastic tissues and that high rate of L-serine synthesis plays a crucial role in invasiveness, malignant transformation, and growth of certain types of human tumors, such as colon carcinoma, breast cancer, and melanoma [77,78]. It has been shown that protein levels of 3-PG dehydrogenase are elevated in 70% of estrogen receptor-negative breast cancers and that suppression of the enzyme in cell lines with its elevated expression causes a strong decrease in cell proliferation [79]. Several studies reported up-regulation of serine hydroxymethyltransferase that catalyzes conversion of L-serine and THF to glycine and N^5^,N^10^-CH_2_-THF [77,80] and contributes to glutathione biosynthesis, the availability of one-carbon groups for synthesis of purines and pyrimidines, and methylation reactions. Furthermore, it has been shown that dietary restriction of L-serine could lower L-serine levels and inhibit tumor growth in mice [81]. Hence, dietary L-serine restriction and serine synthesis inhibitors have attracted wide attention as an emerging therapeutic option in tumor therapy [82].

### 6.7. Serine and Hyperhomocysteinemia

It is believed that hyperhomocysteinemia (blood homocysteine concentration higher than 15 µmol/L) is causally related to an increased risk of atherosclerosis and arterial thrombosis in certain diseases, such as chronic renal failure, hypothyroidism, and tumors. The usual treatment is combined administration of folic acid and B_6_- and B_12_-vitamins.

It is assumed that supplying L-serine will enhance fluxes of homocysteine through the methionine cycle and transsulfuration pathway (Section 5.2 and Section 5.3), which may subsequently reduce homocysteine levels and have therapeutic utility. L-serine decreased homocysteine synthesis in rat hepatocytes incubated with 1 mM methionine [83], reduced plasma homocysteine levels in hyperhomocysteinemia induced by high methionine diet in both rats and humans [84,85,86], and decreased homocysteine levels in mice with alcoholic fatty liver [87].

## 7. L-Serine as a Conditionally Essential Amino Acid

CEAAs are nonessential (dispensable) amino acids that become essential (indispensable) under specific conditions in which the needs for these amino acids are greater than the body’s ability to produce them. As a result of these needs, the exogenous supply of these amino acids must be increased to avoid their deficiency. The best-known examples are glutamine and tyrosine, which are recognized as CEAA in sepsis and chronic renal insufficiency, respectively.

In previous sections of this article, it was shown that dietary intake of L-serine and the body’s ability to ensure whole body serine homeostasis are not sufficient in all circumstances and that L-serine deficiency is associated with altered metabolism and functions, especially of the nervous system. Therefore, L-serine supplementation has been suggested as a rational therapeutical approach in several disorders, particularly primary disorders of L-serine synthesis, neurodegenerative disorders, and diabetic neuropathy. Unfortunately, the number of clinical studies evaluating dietary supplementation of L-serine as a possible therapy is small (Table 1). Moreover, studies examining the therapeutic effects of L-serine in CNS injury and chronic renal diseases, in which it is supposed that L-serine weakens glutamate neurotoxicity and lowers homocysteine levels, respectively, are missing [33,40].

## 8. Conclusions

The results of the studies referred to in this article clearly demonstrate that L-serine should be included in the group of CEAA. The main reasons are as follows:There are primary disorders of L-serine synthesis resulting in L-serine deficiency.Humans cannot synthesize L-serine in sufficient quantities in diabetes and chronic kidney diseases.L-serine deficiency is associated with severe neurological abnormalities.It was proved that L-serine supplementation is effective in therapy of primary disorders of serine metabolism and diabetic neuropathy.

In conclusion, L-serine appears to have a promising therapeutic potential in disorders associated with L-serine deficiency. The mechanisms of potential benefits of increased external supply of L-serine include increased synthesis of SL, decreased synthesis of DSL, decrease in homocysteine levels, and increased synthesis of cysteine and its metabolites, including glutathione. However, more work should be done, including randomized controlled intervention trials in humans utilizing L-serine, to validate its effectiveness for specific disorders.

## Figures and Tables

**Figure 1 nutrients-14-01987-f001:**
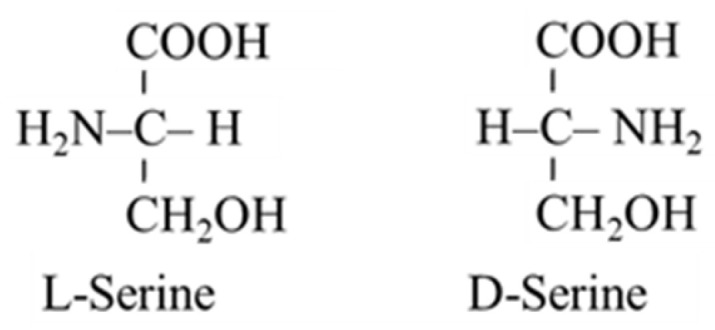
Chemical structures of L-serine and D-serine.

**Figure 2 nutrients-14-01987-f002:**
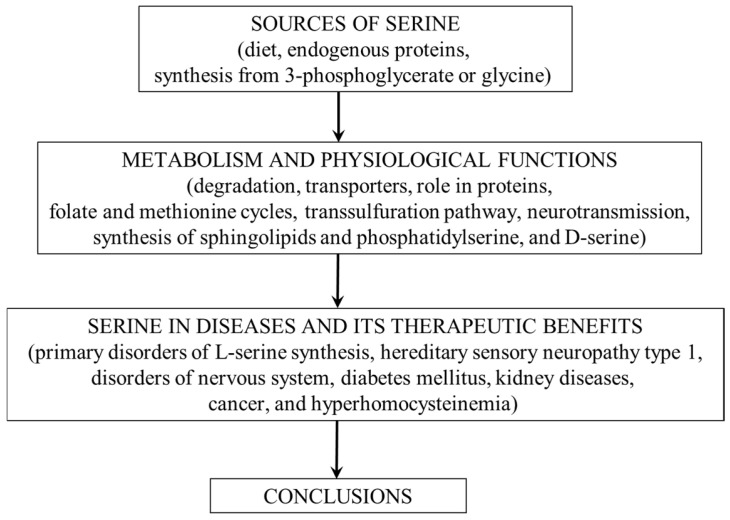
The conceptual framework of the article.

**Figure 3 nutrients-14-01987-f003:**
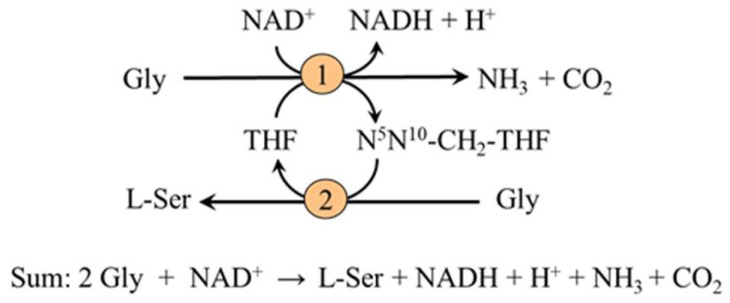
L-serine synthesis from glycine in the kidneys. 1, glycine cleavage enzyme; 2, serine hydroxymethyltransferase.

**Figure 4 nutrients-14-01987-f004:**
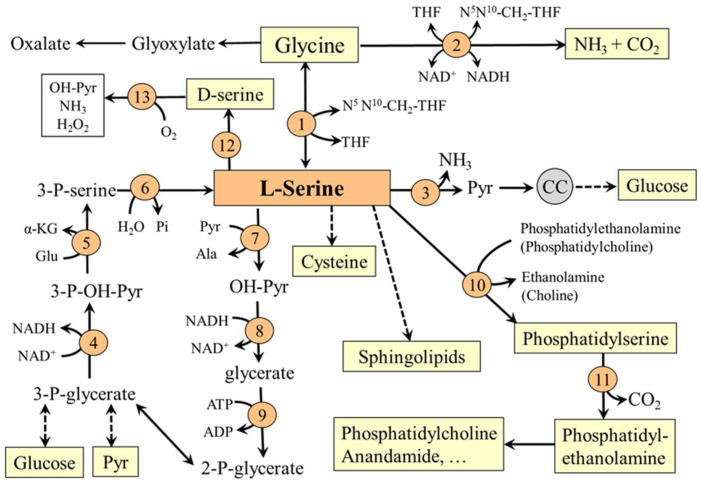
Pathways of serine synthesis and metabolism. 1, serine hydroxymethyltransferase; 2, glycine cleavage system; 3, serine dehydratase; 4, 3-phosphoglycerate dehydrogenase; 5, phosphoserine aminotrasferase; 6, phosphoserine phosphatase; 7, serine-pyruvate aminotransferase; 8, glycerate dehydrogenase; 9, glycerate kinase; 10, phosphatidylserine synthase; 11, phosphatidylserine decarboxylase; 12, racemase; 13, D-amino acid oxidase. CC, citric cycle; Pyr, pyruvate.

**Figure 5 nutrients-14-01987-f005:**
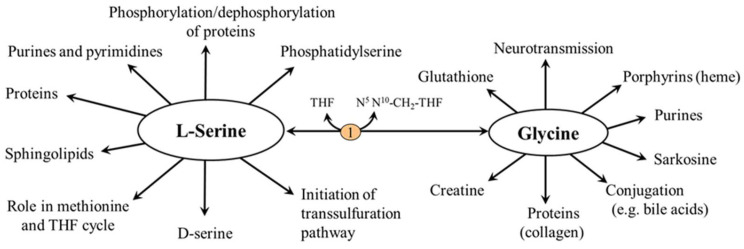
Main functions of L-serine. 1, serine hydroxymethyltransferase.

**Figure 6 nutrients-14-01987-f006:**
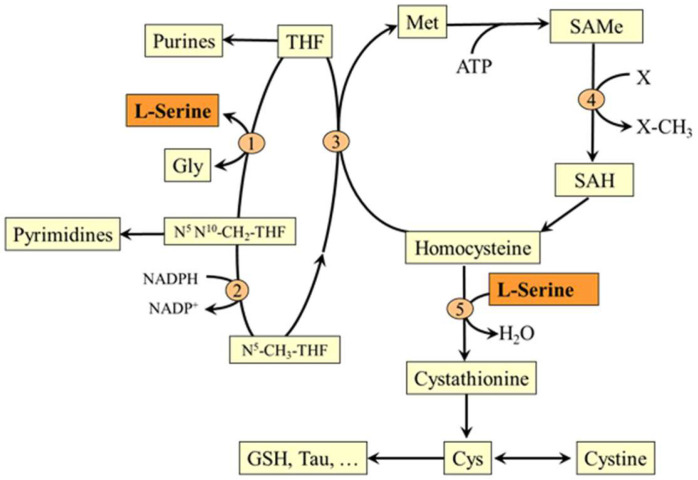
Relationship of L-serine to folate and the methionine cycle and transsulfuration pathway. 1, serine hydroxymethyltransferase; 2, methylene tetrahydrofolate reductase; 3, methionine synthase; 4, methyltransferase; 5, cystathionine β-synthase. SAH, S-adenosylhomocysteine; SAMe, S-adenosylmethionine. GSH, glutathione.

**Figure 7 nutrients-14-01987-f007:**
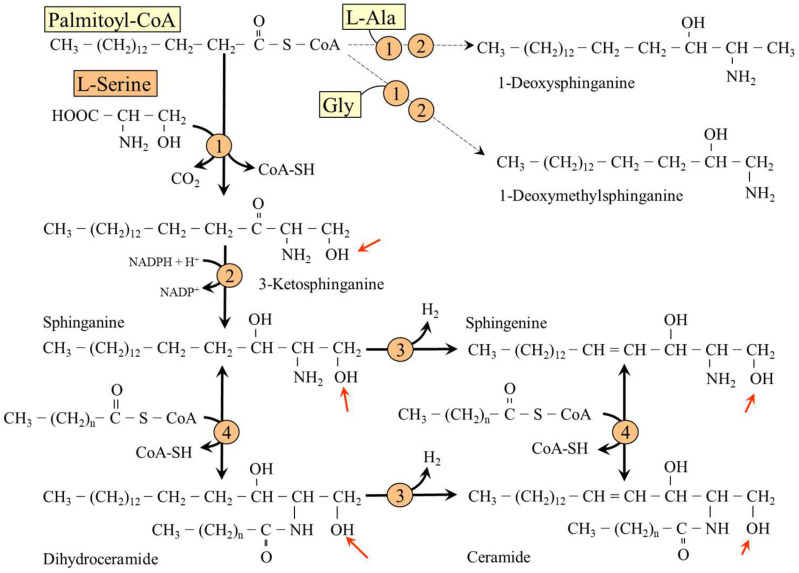
Synthesis of sphingosines, ceramides and deoxysphingosines. 1, Serine palmitoyltransferase; 2, ketosphinganinereductase; 3, dehydrogenase; 4, ceramidesynthase. Arrows indicate a hydroxyl group used for the synthesis of various types of SL that is missing in DSL.

**Figure 8 nutrients-14-01987-f008:**
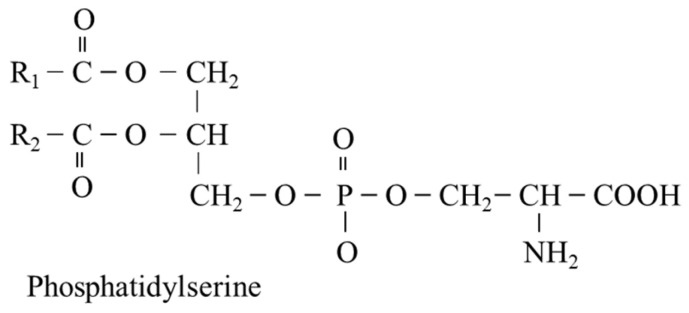
Chemical structure of phosphatidylserine. The fatty acid at position 1 is saturated (R1), the fatty acid at position 2 is unsaturated (R2).

**Table 1 nutrients-14-01987-t001:** Reports of therapeutic benefits of L-serine in humans.

Disease	Reference(s)
Primary disorders of serine synthesis	[6,7,54,55]
Hereditary sensory neuropathy type 1	[2,56]
Amyotrophic lateral sclerosis	[8,88]
Encephalopathy due to mutations of NMDA receptor	[89]
Diabetes	[4,9]
Hyperhomocysteinemia	[86]
Chronic renal injury	None
CNS injury	None

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
