# Peer review of "Serine Metabolism in Health and Disease and as a Conditionally Essential Amino Acid"

_nutrients, 2022, doi:10.3390/nu14091987_

Round 1

Reviewer 1 Report

Milan Holeček reviews the evidence supporting L-serine as a conditionally essential amino acid. By describing studies which advocate for the requirements of L-serine and highlighting its potential therapeutic use in some disorders, Holeček's review covers several key health aspects of serine metabolism. Additional details and studies describing the mechanistic pathways and molecular mechanisms by which serine is having a  therapeutic effect in specific disorders would strengthen the review even further.

Reviewer 2 Report

It is a good piece of well written review. But one important critical element that is missing in this manuscript is, the conceptual frame work. Authros are requested to incorporate to synthesize the conceptual frame work, incorporate the same into the manuscript and add an explanation of the same in detail.

Author Response

Thank you very much for the positive evaluation of my article.

I have incorporated a scheme of conceptual framework into the manuscript (please see Figure 2).

Reviewer 3 Report

This review by Holecek focuses on the role of an important amino acid, L-serine in various metabolic and cellular processes.

The manuscript is well-organized and well-written, and covers all aspects of serine metabolism, and its role in various diseases. References are appropriately cited. 

There are several reviews already published on L-serine in neurological diseases, and so the focus on other diseases involving serine is an important aspect of this review.  

Author Response

Thank you very much for nice evaluation of my article.